# Modification of Carbon Black with Hydrogen Peroxide for High Performance Anode Catalyst of Direct Methanol Fuel Cells

**DOI:** 10.3390/ma14143902

**Published:** 2021-07-13

**Authors:** Yu-Wen Chen, Han-Gen Chen, Man-Yin Lo, Yan-Chih Chen

**Affiliations:** 1Department of Chemical Engineering, National Central University, Jhongli 32001, Taiwan; tsubomi0915@gmail.com; 2Department of Chemical Engineering, Material and Chemical Research Laboratories, Industrial Technology Research Institute, Hsinchu 30041, Taiwan; manyin@gmail.com (M.-Y.L.); yanchen@itri.org.tw (Y.-C.C.)

**Keywords:** carbon black, fuel cell, PtAu catalyst, anode catalyst, direct methanol fuel cell

## Abstract

In this study, high-surface-area carbon black is used to support PtRu. In order to increase the functional groups on the surface of carbon black and to have a more homogenous dispersed PtRu metal, the surface of carbon black is functionalized by H_2_O_2_. PtRu/carbon black is synthesized by the deposition–precipitation method. NaH_2_PO_2_ is used as the reducing agent in preparation. These catalysts are characterized by N_2_ sorption, temperature-programmed desorption, X-ray diffraction, transmission electron microscope, and X-ray photoelectron spectroscopy. The methanol oxidation ability of the catalyst is tested by cyclic voltammetry measurement. Using H_2_O_2_ to modify carbon black can increase the amount of functional groups on the surface, thereby increasing the metal dispersion and decreasing metal particle size. NaH_2_PO_2_ as a reducing agent can suppress the growth of metal particles. The best modified carbon black catalyst is the one modified with 30% H_2_O_2_. The methanol oxidation activity of the catalyst is mainly related to the particle size of PtRu metal, instead of the surface area and conductivity of carbon black. The PtRu catalyst supported by this modified carbon black has very high activity, with an activity reaching 309.5 A/g.

## 1. Introduction

In the last decades, direct methanol fuel cells (DMFC) have gained attention due to their zero-pollutant emission, high efficiency, fast start up time and low operating temperature [1,2]. PtRu is used as the catalyst extensively. Since a very high PtRu metal loading is used, the particle size of PtRu is large. Achieving a small particle size of PtRu is an emergent issue in application [3,4,5,6,7,8,9,10,11,12,13,14,15].

Carbon black has been used extensively as the anode of fuel cells, with very few functional groups on the surface, as well as a low PtRu metal dispersion and low metal-support interaction, due to the small amount of function groups on its surface. This results in the low activity of fuel cells. Various oxidants, such as H_2_O_2_ [9,10], HNO_3_ [11], NaOH [12] and O_3_ [15] have been used to modify the surface of carbon black to create more function groups on the surface [16,17,18]. It is believed that this will result in a high metal-support interaction and high metal dispersion. However, modification of carbon black may result in low conductivity, and the treatment condition remains unclear. Furthermore, an appropriate treatment condition is needed to have high PtRu metal dispersion without losing too much conductivity.

In the present work, platinum-ruthenium metal supported on modified carbon black has been successfully prepared. Carbon black (Ketjen black ECP300), which has a high surface area (855 m^2^/g), was used as the substrate. H_2_O_2_ was used to modify the surface of carbon black to have more functional groups, which may increase metal-support interaction and results in a high metal dispersion. The results indicate that a uniform dispersion of PtRu alloy nanoparticles in modified carbon black is achieved. Moreover, single-cell measurements revealed that the prepared electrocatalyst had a high oxygen reduction reaction (ORR) activity.

## 2. Experimental

### 2.1. Chemicals

Hydrogen peroxide was from Showa Chemicals (Tokyo, Japan). Chloroplatinic acid hydrate was from Aldrich (Saint Louis, MO, USA). Ruthenium chloride hydrate was sourced from Sigma-Aldrich (Darmstadt, Germany), ethanol was from a local vender, Hydrochloric acid and sulfuric acid were from Fisher (Hampton, NH, USA). Nafion solution was from DuPont (Wilmington, DE, USA). Carbon black was from Ketjen (ECP300JD, Tokyo, Japan). Hydrogen and nitrogen were form Air Products (Allentown, PA, USA).

### 2.2. Preparation of the Anode Catalyst

In this study, ECP300 carbon black from Ketjen (Tokyo, Japan) was used as the substrate. Then, 10 g of carbon black was added in water containing various amounts of H_2_O_2_. This was heated at 95 °C for 24 h. The sample was filtered and washed with distilled water and then dried at 110 °C for 3 h.

PtRu/carbon black was synthesized by the deposition–precipitation method. RuCl_3_ and H_2_PtCl_6_ were dissolved in distilled water, carbon black support was then added in the above solution. It was agitated by supersonic vibration for 1 h. The reducing agent NaH_2_PO_2_ was added to the above solution. The sample was washed in distilled water and dried in a vacuum oven. The pretreatment conditions of the samples are listed in Table 1.

### 2.3. Characterization of Catalyst

#### 2.3.1. XRD

The XRD pattern of each sample was recorded on a diffractometer (PANALY X’PERT, Philips, Amsterdam, The Netherlands) with Cu-K_α_ radiation. The sample was scanned from 10° to 85° at a scanning rate of 0.025°/s.

#### 2.3.2. N_2_ Sorption

The surface area (S_g_), pore volume (V_p_), and pore distribution of catalysts were measured by ASAP2400 (Micromeritics Instrument Corporation, Norcross, GA, USA) surface area and pore size analyzer. N_2_ adsorption–desorption isotherms were obtained at −196 °C by the BET (Brunauer-Emmett-Teller) equation for surface area and the BJH (Barrett, Joyner, and Halenda) method for pore size distribution.

#### 2.3.3. Temperature-Programmed Desorption (TPD)

TPD was used to monitor the oxygen-containing functional groups on the surface of carbon black. At high temperatures, the functional groups would decompose to release CO and CO_2_. Based on the TPD peaks, one can determine the acidity of the functional group on the surface.

First, 0.5 g of the sample was loaded in a quartz U tube reactor from a local vender. Helium was used as the carrier gas. The temperature of the reactor was raised to 1000 °C with a heating rate of 5 °C/min. The outlet gas was sampled every 1 min and analyzed by a gas chromatograph (China Gas Chromatography Company, Taiwan) with a TCD detector (Shimadzu, Tokyo, Japan).

#### 2.3.4. TEM

Transmission electron micrographs (TEM), energy dispersion scanning (EDS) and selective area electronic diffraction (SAED) analyses were performed on a JEM-2000FX (Berkeley, CA, USA) instrument using an accelerating voltage of 200 kV.

#### 2.3.5. X-ray Photoelectron Spectroscopy (XPS)

The XPS spectra were recorded with an ESCALAB 250 XPS (VG Scientific, Waltham, MA, USA). The XPS spectra were collected using Al K_α_ radiation at a voltage and current of 20 kV and 30 mA, respectively. The base pressure in the analyzing chamber was maintained in the order of 10^−9^ Pa. The pass energy was 23.5 eV and the binding energy was calibrated by contaminant carbon (C 1s = 284.5 eV). The peaks of each spectrum were organized using XPSPEAK4.1 software (Informer Technologies, Inc., Los Angeles, CA, USA); Shirley type background and 30:70 Lorentzian/Gaussian peak shape were adopted during the deconvolution.

#### 2.3.6. X-ray Fluorescence (XRF)

An XRF (Philips PW2400 Spectrometer, Amsterdam, The Netherlands) was used to analyze the compositions of the samples.

#### 2.3.7. Conductivity of Carbon Black

The conductivity of the carbon black was measured with four-point Probe Tester and Potentiostat (Autolab PGSTAT30, Utrecht, The Netherlands).

### 2.4. Methanol Oxidation Activity

The methanol oxidation activity of PtRu/carbon black was measured with a half cell.

#### 2.4.1. Preparation of Working Electrode

10 mg of PdRu/modified carbon black catalyst was added in 6 mL distilled water. It was stirred with a supersonic vibrator for 30 min until completely mixed. A 20 μL catalyst slurry was dropped on the rotating electrode disk. It was dried at 50 °C for 50 min, then cooled to room temperature in a nitrogen atmosphere. 10 μL (1%) Nafion in IPA solution was then added in dry catalyst. It was cooled in a nitrogen atmosphere until the Nafion was dried.

#### 2.4.2. Cyclic Voltammetry

Catalyst ink was prepared by dispersing 40 mg of the electrocatalyst in 20 mL methanol and was subjected to ultrasonification for 30 min. Then, 20 μL of the catalyst solution was dropped onto a glassy carbon surface with an exposed area of 0.196 cm^2^. After drying the droplet at 60 °C, 20 μL of 1 wt.% Nafion alcoholic solution was dropped on the electrode surface and heated at 60 °C. The electrode was pretreated to remove surface contamination by cycling the electrode potential between 0 and 1.0 V vs. RHE at 50 mV s^−1^ for 100 cycles in 0.5 M H_2_SO_4_. Electrocatalytic methanol oxidation was then measured by a chronoamperometry in 1 M CH_3_OH mixed with 0.5 M H_2_SO_4_. The current was recorded at 1800 s after stepping the potential to 0.50 V, which was used as the quasi-steady-state current. Catalytic activity was calculated by the current density at 0.5 V obtained in CV method divided by PtRu catalyst loading.

#### 2.4.3. Statistical Analysis

All experiments were run in triplicate and the statistical significance was analyzed by using SPSS statistical program (version 20, Armonk, NY, USA). Data were subjected to one-way analysis of variance (ANOVA), and Duncan’s multiple range tests were performed for mean comparison. Significance was defined at *p* values less than 0.05.

## 3. Results and Discussion

### 3.1. Characterization of the Modified Carbon Black

Carbon black is a very good electrocatalyst since it has very high conductivity and stability [13,14,15,16,17,18,19]. However, it lacks functional groups on the surface, resulting in low metal-support interaction and low metal dispersion and large metal particle size. This is the main reason to have low activity in DMFC. One of the methods to increase the amount of functional groups on the surface of carbon black is to oxidize the surface of carbon black. This can increase the hydrophilicity of carbon black, to enhance the PtRu metal-carbon black interaction, resulting in a small particle size of PtRu metal and a high activity of the catalyst.

In this study, H_2_O_2_ was used to modify the surface of carbon black. The effects of H_2_O_2_ modification on the functional groups of the surface of carbon black and the conductivity and activity of PtRu/carbon black catalyst were investigated. N_2_ sorption was used to measure the BET surface area and pore size distribution of the catalyst. The acidity of surface of the catalyst was measured by temperature-programmed desorption of functional groups. The CO desorption related to weak acid sites, and CO_2_ desorption corresponded to strong acid sites. The strong acid sites include lactones, anhydride, and carboxylic acid functional groups. The weak acid sites include carbonyl, quinine, and phenol.

#### 3.1.1. TPD of Functional Groups on the Surface of Carbon Black

Figure 1 shows only a very small amount of oxygen-containing functional groups in the unmodified carbon black. If the surface of carbon black has a very small amount of functional groups, it would result in low metal dispersion. After H_2_O_2_-modification, carbon black had more functional groups on the surface. The TPD peak at low temperatures was mainly from the strong acidic functional groups, and those which desorbed at high temperatures resulted from weak acidic functional groups. The H_2_O_2_-treated carbon black showed a high concentration of weak acid sites, but a low concentration of strong acid sites according to the TPD results. It should be noted that the weak acid sites are the sites to deposit metals.

#### 3.1.2. BET Surface Area and Pore Size Distribution of Modified Carbon Black

The BET surface area was obtained by N_2_ sorption isotherm and pore size distribution was calculated by BJH method. Table 2 shows the BET surface areas and pore structure of the modified carbon black. The surface area of the unmodified carbon black was 855 m^2^/g. The surface area decreased after modification by H_2_O_2_. It was also influenced by the amount of H_2_O_2_. If the carbon black was modified with 150 mL H_2_O_2_, the surface area decreased significantly. The pore volume of micropores changed slightly after modification. The increase in pore size was also very limited, only changing from 59 to 68 Å. The mesopore volume of the carbon black increased slightly after H_2_O_2_-modification. The results show that the pore structure of carbon black was influenced by H_2_O_2_ modification. The surface of pore in carbon black was oxidized by H_2_O_2_, and thereby the pore structure was changed.

#### 3.1.3. Conductivity of Carbon Black after H_2_O_2_ Modification

The modification of carbon black with H_2_O_2_ not only changed its surface properties, but also changed its bulk properties. The electric conductivity of carbon black is influenced by (i) the bulk structure of carbon black, and (ii) the oxygen-containing functional groups on the surface of carbon black. Figure 2 shows that, under H_2_O_2_ modification, the relationship between the concentration of the functional group of carbon black and conductivity is almost a linear relationship. The lower the amount of functional groups is, the higher the conductivity is. The unmodified carbon black had the highest conductivity among all the samples. The conductivity of carbon black decreased with an increase of concentration of functional groups on the surface, in agreement with the literature data [20,21,22,23,24,25,26,27,28,29,30,31,32,33,34,35]. It should be noted that the electric conductivity of carbon black had no correlation with pore structure in this study, which was also in agreement with the literature [27].

It is known that the amount of functional groups on the surface influence the conductivity of carbon black [22,23,24,25,26,27,28,29,30,31,32,33,34,35]. Table 3 shows that the carbon black after H_2_O_2_ modification with 150 mL H_2_O_2_ had higher amount of functional groups (1.653 mmol/g) than those with H_2_O_2_ modified with 100 mL H_2_O_2_. The sample carbon-02 had lower conductivity (5.9 S/cm) than carbon-01. The unmodified carbon black had the higher conductivity and lower amount of functional groups than modified samples.

### 3.2. Characterization of PtRu/Carbon Black Anode Catalyst

PtRu/black carbon was synthesized by the deposition–precipitation method. They were reduced by NaH_2_PO_2_.

#### 3.2.1. XRD

Pt is a face-centered cubic crystal. It has XRD peaks at 39.9°, 46.3° and 67.45°, corresponding to face (111), (200) and (220), respectively [20,21,22,23,24]. Since the atomic radius of Ru is smaller than that of Pt, the spacing of PtRu alloy crystal is smaller than that of Pt, resulting in the shift of XRD peaks to a higher degree. Figure 3 shows the XRD patterns of the PtRu catalysts supported by the modified carbon black and reduced by NaH_2_PO_2_. Using Scherrer’s equation, we were able to calculate the crystallite size of the sample based on the peak intensity of phase (220). We were not able, however, to calculate the crystallite sizes of the sample PtRu/carbon-02, since the peak intensity of Pt (220) was too weak. PtRu/carbon sample had intense XRD peaks, indicating it has large metal particles. Instead, PtRu/carbon-01 sample had a broad Pt (220) peak, indicating small metal particles.

#### 3.2.2. TEM

In this study, carbon black was modified by H_2_O_2_ to have functional groups to increase metal-support interaction.

Table 4 shows the relationship of PtRu particle size with oxygen-containing functional groups on the surface of carbon black. It has been reported [13,14] that metal particle size is related to the functional groups on the surface of carbon black, providing evidence that the catalyst was reduced by NaH_2_PO_2_. As the number of weak acidic functional groups increases, the metal precursors-support interaction increases, which can suppress the agglomeration of metal. PtRu/carbon had a large particle size, and metal particle size distribution was not homogeneous, due to small amount of functional groups on the surface of carbon black, as shown in Figure 4a. PtRu/carbon-01 and PtRu/carbon-02 had very homogeneous metal distribution and small particles, as shown in Figure 4b,c. The results demonstrate that metal dispersion increased with an increase in functional groups on the surface of the support.

Table 4 shows that the amount of functional groups increased and metal particle size decreased with an increase in the amount of H_2_O_2_ during modification. Table 4 shows that as the functional groups on the surface of the carbon black increased, the amount of anchored sites for metal increased, and the metal particle size decreased. The results confirm that the surface modification can increase the amount of functional groups, increase metal dispersion, and decrease metal particle size. In conclusion, by using carbon black containing a high concentration of functional groups and using NaH_2_PO_2_ as the reducing agent, one is able to obtain small PtRu metal particles.

H_2_O_2_-treated carbon black shows a high concentration of weak acid sites, and a low concentration of strong acid sites according to the TPD results. H_2_PtCl_6_ shows a strong interaction of the metallic precursor with the carbon black of low acidity. Carbon black functionalized with H_2_O_2_, which develops acidic sites with moderate strength and shows a strong interaction with H_2_PtCl_6_ during the deposition–precipitation process, would favor the PtRu dispersion on the surface of carbon black.

#### 3.2.3. XPS

In the literature [35], PtRu-P had interactions among three elements, and the binding energy of Pt 4f became higher. The theoretical value of the binding energy of Pt 4f is 71.1 eV [36,37,38,39,40,41,42]. Since Pt should form an alloy with Ru to prevent poisoning, the electronic state of Ru was investigated. Most researchers used Ru 3p to examine the electronic state of Ru, because Ru 3d’s peak overlaps with C 1s. The peak intensity of Ru 3p was weak, and the noise was significant. In this study, Ru 3d peaks were studied (Figure 5 and Figure 6). In Table 5, all peak intensities were divided by ASF (atomic sensitivity factor), and C 1’s peak intensity was used as a basis for comparison. The results show that sample PtRu/carbon had the least amount of Pt^0^ among all samples, PtRu/carbon-02 had the second least, PtRu/carbon-01 sample had the highest amount of Pt^0^. The PtRu/carbon-01 sample also had the highest amount of Pt-RuO_2_ on the surface among all samples, containing 1.57% Pt and 1.33% RuO_2_.

### 3.3. Catalytic Activity of the PtRu Catalyst

The activity of the anode catalyst was measured in the half cell with a solution of 100 mL 0.5 M H_2_SO_4_ and 100 mL 1M MeOH. The current density of each catalyst at 0.5 V was used for comparison. The CV plots of the samples are shown in Figure 7.

The unmodified and modified carbon black were used as the supports. 60 wt.% PtRu/carbon black was reduced with NaH_2_PO_2_. Table 6 shows that the metal particle size of the PtRu/carbon-02 catalyst reduced by NaH_2_PO_2_ was 2.33 nm. The ORR activity was 309.05 A/g, which had the highest activity among all the samples. The PtRu/carbon had a lower activity than those supported by the modified carbon black. The results demonstrate that the activity is mainly related to the PtRu particle size, instead of conductivity and pore structure. Carbon black modified with H_2_O_2_ can generate more acidic sites to anchored metal precursors and results in small metal particle size and high activity in methanol oxidation activity. A commercial catalyst JM-60 from Johnson-Matthey was used for comparison, and its activity was 251.32 A/g. Our catalyst was more active than the JM-60.

## 4. Conclusions

Using H_2_O_2_ to modify carbon black can increase the amount of functional groups on the surface, thereby increasing PtRu metal dispersion and decreasing metal particle size. Carbon black treated by H_2_O_2_ has too many functional groups, resulting in a decrease of conductivity. NaH_2_PO_2_ as a reducing agent can suppress the growth of metal particles. The best modified carbon black catalyst is the one modified with 30% H_2_O_2_. H_2_O_2_-treated carbon black shows a high concentration of weak acid sites but a low concentration of strong acid sites. Carbon black functionalized with H_2_O_2_, which develops acidic sites with moderate strength and shows a strong interaction with metal precursors during preparation, favors PtRu dispersion on the surface of carbon black and consequently the catalytic behavior. The PtRu catalyst supported on this modified carbon black had very high activity, with an activity that can reach 309.5 A/g.

The results demonstrate that this activity is mainly related to the PtRu particle size, instead of conductivity and pore structure. Carbon black modified with H_2_O_2_ can generate more acidic sites to anchored metal precursors, and results in small metal particle size as well as a high methanol oxidation activity.

Further research on various modified agents, such as O_3_ and HNO_3_, as well as various reducing agents, are needed to improve the activity of the catalyst.

## Figures and Tables

**Figure 1 materials-14-03902-f001:**
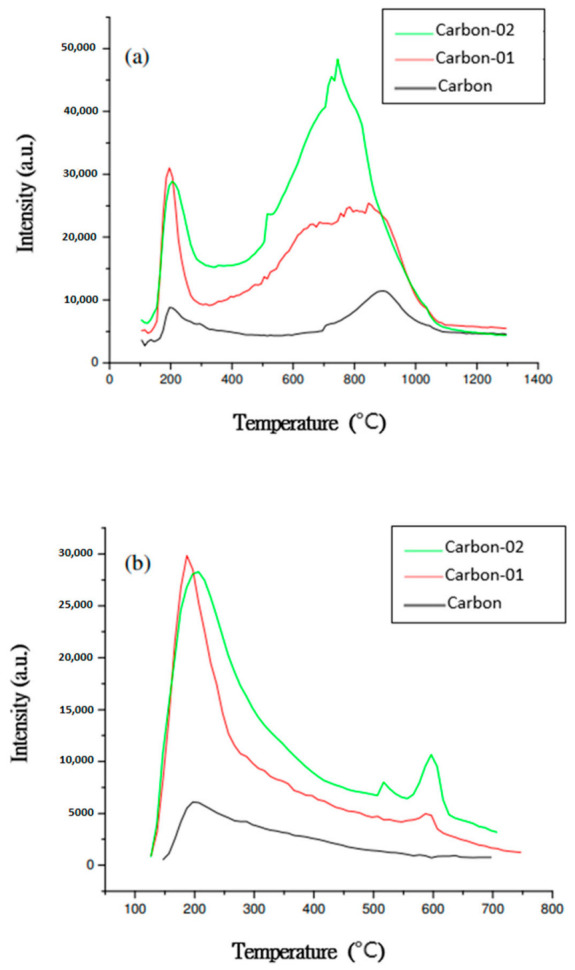
TPD profile of CO and CO_2_ on (**a**) CO desorption, (**b**) CO_2_ desorption.

**Figure 2 materials-14-03902-f002:**
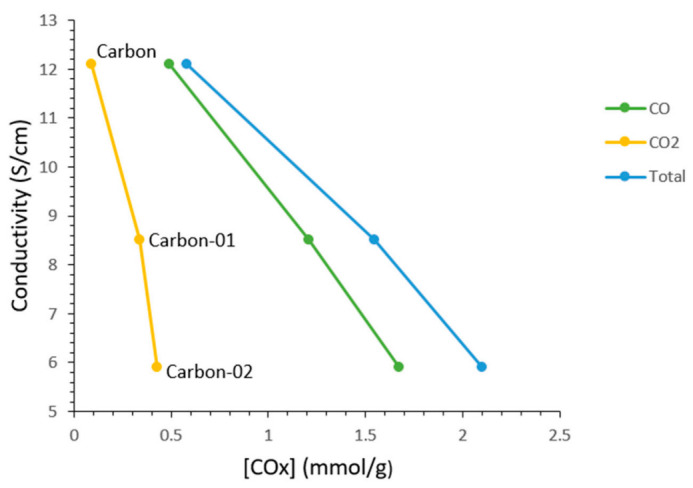
Conductivity vs. concentration of oxygen-containing functional groups of carbon black under various modification conditions. carbon-01: modified with 100 mL H_2_O_2_; carbon-02: modified with 150 mL H_2_O_2_.

**Figure 3 materials-14-03902-f003:**
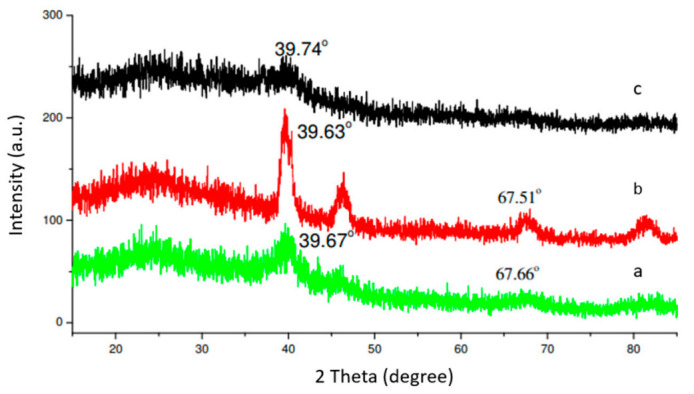
XRD patterns of 60 wt.% PtRu supported on H_2_O_2_-modified carbon black and reduced by NaH_2_PO_2_. (**a**): PtRu/carbon; (**b**): PtRu/carbon-01; (**c**): PtRu/carbon-02.

**Figure 4 materials-14-03902-f004:**
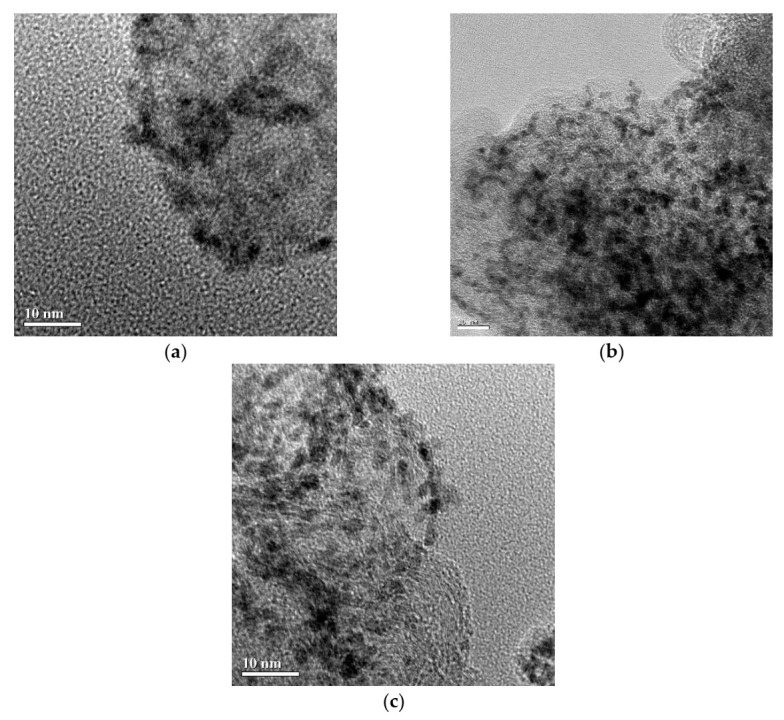
TEM images of the catalyst which was modified by H_2_O_2._and reduced by NaH_2_PO_2_. (**a**) PtRu/carbon; (**b**) PtRu/carbon-01; (**c**) PtRu/carbon-02.

**Figure 5 materials-14-03902-f005:**
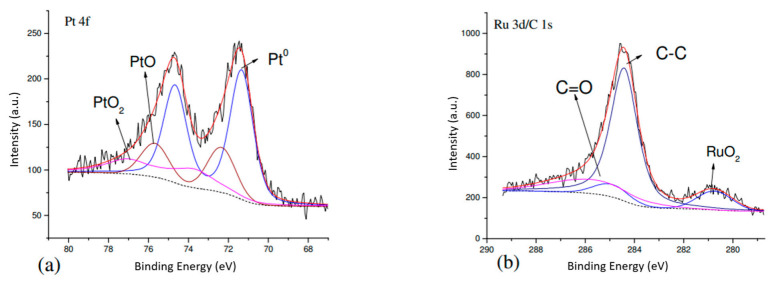
XPS spectra of PtRu/carbon-01. (**a**) Pt 4f spectra; (**b**) Ru 3d/ C 1s spectra.

**Figure 6 materials-14-03902-f006:**
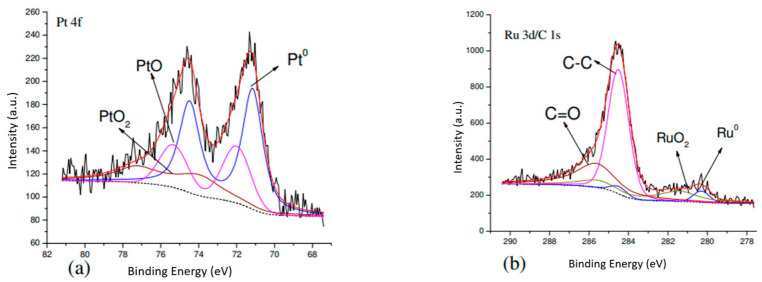
XPS spectrua of PtRu/carbon-02. (**a**) Ru 3d/C 1s spectrum, (**b**) Pt 4f spectrum.

**Figure 7 materials-14-03902-f007:**
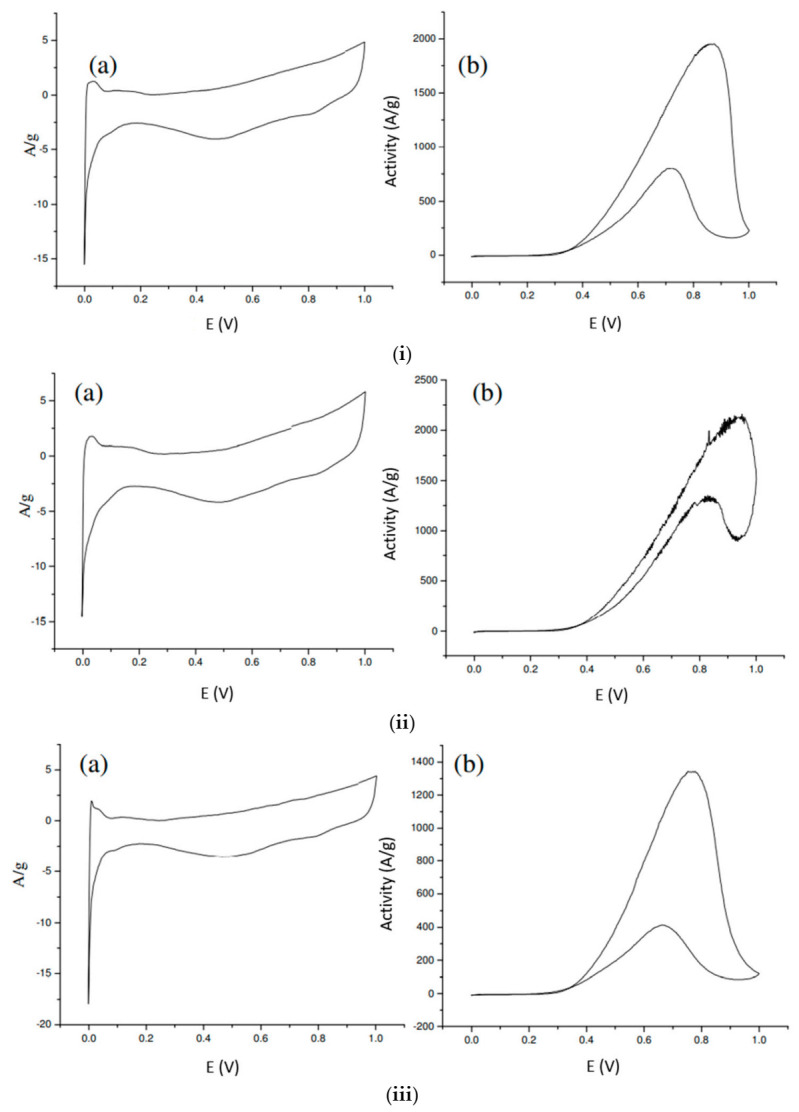
CV plot of the sample. (**i**) Sample PtRu/carbon-01; (**ii**) Sample PtRu/carbon-02; (**iii**) Sample PtRu/carbon. (**a**) background of the 10th cycle, (**b**) activity of the 10th cycle.

**Table 1 materials-14-03902-t001:** Pretreatment condition of carbon black (10 g) treated with various amounts of H_2_O_2_.

Amount of 30 wt.% H_2_O_2_	Temperature	Treatment Time	Notation of the Sample
	(°C)	(h)	
0			carbon
100 mL	95	24	carbon-01
150 mL	95	24	carbon-02

**Table 2 materials-14-03902-t002:** Modification condition of carbon black and their pore structure.

Sample	Modification Condition			Surface Area (m^2^/g)	Micropore Volume (cm^3^/g)	Micropore Area (cm^2^/g)	Total Pore Volume (cm^3^/g)	Pore Diameter (Å)
	Agent	Time (h)	T (°C)					
				BET	t-Method	t-Method	BET	BJH
carbon	-	-	-	855	0.058	142	1.25	59
carbon-01	100 mL 30% H_2_O_2_	24	95	830	0.054	132	1.41	68
carbon-02	150 mL 30% H_2_O_2_	24	95	722	0.052	123	1.09	61

**Table 3 materials-14-03902-t003:** Effect of H_2_O_2_ treatment on the conductivity of the modified carbon black.

Sample	[O] (mmol/g)			Conductivity (S/cm)
	CO	CO_2_	Total	
carbon	0.4915	0.0905	0.582	12.11
carbon-01	1.2074	0.3391	1.546	8.50
carbon-02	1.6718	0.4267	2.098	5.90

**Table 4 materials-14-03902-t004:** Pt particle size of the sample supported by H_2_O_2_-modified carbon black and reduced by NaH_2_PO_2_.

Sample	[O] (mmol/g)			Metal Particle Size (nm)
	CO	CO_2_	Total	TEM
PtRu/carbon	0.492	0.0901	0.58	2.59
PtRu/carbon-01	1.207	0.339	1.55	2.48
PtRu/carbon-02	1.672	0.427	2.10	2.33

**Table 5 materials-14-03902-t005:** Surface compositions of the catalysts.

Sample	Species	Relative Concentration	Mole Ratio *	BE ** (eV)	FWHM	NIST XPS Database	
						BE (eV)	Formula
PtRu/carbon	Pt 4f			4f7/2		4f7/2	
		53.79%	0.68%	71.2	1.38	71.1	Pt^0^
		46.21%	0.59%	72.5	2.38	72.4	PtO
	Ru 3d			3d5/2		3d5/2	
		100%	1.90%	282	1.82	281	RuO_2_
	C 1s					1s	
				284	1.31	284.5	C−C
				286	4.55	285.85	−C=OH
PtRu/carbon-01	Pt 4f			4f7/2		4f7/2	
		60.04%	1.57%	71.3	1.37	71.1	Pt^0^
		22.00%	0.58%	72.3	1.64	72.4	PtO
		17.96%	0.47%	73.8	2.54	74.1	PtO_2_
	Ru 3d			3d_5/2_		3d_5/2_	
				280	0.74	280.1	Ru^0^
		100%	1.33%	281	1.63	281	RuO_2_
	C 1s					1s	
				284	1.29	284.5	C−C
				286	4.01	285.85	−C=OH
PtRu/carbon-02	Pt 4f			4f7/2		4f7/2	
		53.50%	1.16%	71.2	1.28	71.1	Pt^0^
		26.66%	0.58%	72	1.75	72.4	PtO
		19.85%	0.43%	73.9	2.65	74.1	PtO_2_
	Ru 3d			3d5/2		3d5/2	
		24.55%	0.50%	280	0.96	280.1	Ru^0^
		75.45%	1.53%	281	2.27	281	RuO_2_
	C 1s					1s	
				285	1.24	284.5	C−C
				286	2.42	285.85	−C=OH

* Obtained from peak intensity divided by ASF (Pt 4f = 5.575, Ru 3d = 4.273, C 1s = 0.296) ** BE: Binding energy.

**Table 6 materials-14-03902-t006:** Effects of carbon black modified with various amounts of H_2_O_2_ on the activity of the catalysts.

Catalyst Notation	[O] (mmol/g)			Conductivity of Carbon (S/cm)	Metal Particle Size (nm)	XRF (at%)		Loading (wt.%)		Activity (A/g)
	CO	CO_2_	Total		TEM	Pt	Ru	Theo	TGA	
PtRu/carbon	0.492	0.0901	0.58	12.11	2.59	57	43	59.88	55.02	253.2
PtRu/ carbon-01 *	1.207	0.339	1.55	8.50	2.48	60	40	59.93	55.70	295.0
PtRu/ carbon-02 *	1.672	0.427	2.10	7.62	2.33	58	42	59.49	56.43	309.5

* Catalyst was prepared at 95 °C.

## Data Availability

Data sharing is not applicable.

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
