# Peer review of "Modification of Carbon Black with Hydrogen Peroxide for High Performance Anode Catalyst of Direct Methanol Fuel Cells"

_materials, 2021, doi:10.3390/ma14143902_

Round 1

Reviewer 1 Report

The manuscript describes a modification of Carbon Black with hydrogen peroxide for high-performance anode catalyst of the direct methanol fuel cell. The manuscript presented concerns an interesting and actual subject. This manuscript can be accepted after major revision.

Some comments are listed below:

  1. The authors could insert more numerical data into the Abstract.
  2. The introduction is rather short. Please add some information about others carbons modified with hydrogen peroxide. Please cite (1) Electrocatalysis of Hydrogen Peroxide Generation Using Oxygen-Fed Gas Diffusion Electrodes Made of Carbon Black Modified with Quinone Compounds. Electrocatalysis 11, 338–346 (2020). https://doi.org/10.1007/s12678-020-00591-1. (2) Effective Synthesis of Carbon Hybrid Materials Containing Oligothiophene Dyes, Materials 12 (20), 3354, 2019 https://doi.org/10.3390/ma12203354. (3) The role of activated carbons functionalized with thiol and sulfonic acid groups in catalytic wet peroxide oxidation, Applied Catalysis B: Environmental, 106, 3-4, 390-397, 2011 https://doi.org/10.1016/j.apcatb.2011.05.044.
  3. Please seek guidance from a native English speaker if possible (some words, commas, "the", etc.).
  4. Why authors use only the BJH method, not others methods to comparison? Can the authors explain that?
  5. Can the authors add figure with the isotherms?
  6. Please increase text in 3.1.2. Add a sentence about the mesopore structure of the authors materials.
  7. Table 2 "Micropore volume, not "micrepore"
  8. Can the authors explain more about the lower conductivity in modified samples?
  9. Please change Figure 4 to better quality. Can the authors add in figure TEM/EDS measurements about the black fragments? Why authors use only one magnification. Can you explain that? Please add different magnification for comparison.
  10. Could the authors include the standard deviation of the statistical analysis in elemental analysis?
  11. Can authors add SEM images?
  12. Please add to the results part, the Raman spectra and ID/IG ratio and comments to the text?
  13. Please add the cyclic voltammetry figure (CV curves).
  14. Authors are suggested to describe some future plans in conclusions and add some sentences for enhancement.

Reviewer 2 Report

This article is impossible to understand. English is poor and redaction is not revised. Figures are cut somehow and it is not possible to follow, since some paragraphs discuss something which is not even present in the figures and viceversa. In addition, samples labbeling is changed all along the manuscript.

Besides the difficulty on following authors' statements, the results are niether properly presented nor discussed. I see no novelty and no discussion on the work presented. Why everything authors are commmenting is happening?

This work has to be rejected and properly prepared if authors want it published in any scientific journal.

Reviewer 3 Report

The MS needs a complete spell and grammar check as the numerous errors obstruct the reading of otherwise important contribution. Drawing and text overlap on various Figures in the MS, making them unreadable. Better discussions and conclusion are needed. Major revision recommended.

Some spell and grammatical errors: "It is believe" (p.1); "wasmeasured" (p.3); "sine" (p.4), "is over-does" (p.10)

10 g ECP 
p.2.  Preparation of anode catalyst is unclear. "10 g ECP 300 carbon black was added in water containing various amounts of H2O2. It was heated at 95 oC for 24 h."  I assume the suspension was heated to remove traces of H2O2 after oxidation of carbon. The temperature and time during the oxidation should be indicated. 

p.3. Preparation of the working electrode is unclear: I believe the sentence: "The sample was dried..." should be removed as the drying procedure is repeated. 

p.4. "Fig.1 shows only very small amount...." As the figure shows well resolved peaks, the sentence is confusing and needs to be clarified with the criterion for comparison, i.e. the amount is small as compared to...

p.4. Needs references for CO and CO2 functional groups and their TPD relationship.

p.5. There is no discussion for the decrease in surface area and pore volume in Table 2. If discussed later in the MS, it should be stated so. 
p.5.   "Figure 2 shows that, under H2O2 modification, the relationship between concentration of functional group of carbon black and conductivity." The sentence is not finished.

p.5. Figure 2. Figure and text are overlapping in Figure 2. The caption talks about carbon 01 and carbon 02, but the picture shows ECP300-07 and -08.

p.6, Figure 3. Overlapping of text and the spectra makes the figure unclear. As in Fig. 2, the caption does not match the samples indicated on the graph.

p.7. "...binding energy of Pt 4f is 71.1 [36-42]." Units are missing.

Reviewer 4 Report

The work of Y-W et al. titled "Modification of Carbon Black with Hydrogen Peroxide for High Performance Anode Catalyst of Direct Methanol Fuel cell" is of interest, authors succeeded to some extent to demonstrate the benefit of functionalizing carbon black to achieve a direct methanol FC, however some concerns need to be corrected.

1- Most of the figures are of a bad quality. Captions are a mess

2- The BET plots are missing and not commented in the manuscript

3- TEM results don't bring any additional information and not exploited to support their claims. There is no particles size measurement or any kind of analysis to help the reader to understand why the authors used TEM

4- Where are the TGA plots and analysis

5- These experiments are missing : XRF; oxidation activitvity plots

6- XRD plots (is it 2theta? )

7- Regarding the TPD experiments: what is the relationship between the hydrophilicity and the O content. Hydrophilicity is not directed linked to element and it a surface property. Authors need to address this question carrefully

Round 2

Reviewer 1 Report

Accept in present form.

Author Response

Thanks a lot for your time and helpful comments. I highly appreciate your comments. I hope to meet you in some conference.

Reviewer 2 Report

I still consider that this paper does not provide any novelty and the results are poorly discussed

Author Response

Thank you very much for your time and comments. I have revised the manuscript. In this study, carbon black was modified with various amounts of H2O2. 60 wt.% PtRu were loaded with deposition-precipitation method, and was reduced by NaH2PO2.  A commercial catalyst from Johnson-Matthey JM-60 was included for comparison. Our results demonstrate that the methanol oxidation activity is strongly related to the metal particle size. PtRu supported on the modified catalyst also had a higher activity than the unmodified one and the commercial catalyst. I hope you can accept my reply. 

Reviewer 3 Report

my comments were addressed

Author Response

Thank you very much for your time and comments. I have revised the manuscript again, and included the comparison with JM-60 from Johnson-Matthey. our catalyst was more active than the commercial one.

Reviewer 4 Report

Authors addressed the concerns, further improvements are required to discuss their results. Not all the authors claims are supported by facts especially TEM results.

Author Response

Thanks a lot for your time and comments. The TEM results show the metal particle sizes were smaller on the modified carbon black than that on the unmodified one.

I also included the comparison with the commercial catalyst JM-60 from Johnson-Matthey. Our catalysts are more active than JM-60.

I have revised the manuscript according to your comments.